# Facilitating Synthesis of FeP/C@CoP Composites as High-Performance Anode Materials for Sodium-Ion Batteries

Tianle Mao, Zheyu Hong, Haoran Ding, Jintang Li *, Yongji Xia, Zhidong Zhou and Guanghui Yue *

College of Materials, Xiamen University, Xiamen 361005, China; zdzhou@xmu.edu.cn (Z.Z.)
* Correspondence: leejt@xmu.edu.cn (J.L.); yuegh@xmu.edu.cn (G.Y.)

**Abstract:** Low-cost, high-capacity sodium-ion batteries can help solve energy shortages and various environmental problems. Transition metal phosphides have a high theoretical capacity and a relatively low redox potential (vs. Na/Na$^+$) and are therefore expected to be used as anodes for sodium-ion batteries. Herein, a heterostructure of a FeP/C@CoP composite with a robust structure, fast charge transfer and abundant active sites was rationally designed and synthesized by growing a Co-ZIFs nanoarray on Fe-MOFs and using a phosphiding process. Using this facilitated and cost-effective method, the FeP/CoP bimetal phosphide heterostructures were uniformly embedded into the carbon matrix, and the capacity and cycle stability were effectively improved. The specific capacity of the FeP/C@CoP was as high as 275.7 mA h g$^{-1}$ at a high current density of 5 A g$^{-1}$, and it was still as high as 321.9 mA h g$^{-1}$ after 800 cycles at a current density of 1 A g$^{-1}$. Cyclic voltammetry was used to perform the kinetic analysis, and it was determined that the FeP/C@CoP exhibited an obvious pseudocapacitive behavior during the charge–discharge process of up to 87.4% at a scan rate of 1 mv s$^{-1}$. This work provides a facilitated method of synthesizing composites that can realize a viable strategy for high-performance energy storage.

**Keywords:** sodium-ion batteries; transition metal phosphides; anode; structure design; pseudocapacitive behavior





## 1. Introduction

Recently, sodium-ion batteries (SIBs) have undergone a major breakthrough in the industry due to the attention of many researchers. They offer significant cost and natural abundance advantages over existing lithium-ion battery systems [1–3]. Sodium-ion batteries could play a key role in the development of more affordable electric vehicles in place of lithium-ion batteries, helping to address environmental impacts. Enhancing the efficiency of anode materials is crucial for facilitating the practical application of SIBs. While the mechanism of sodium-ion storage resembles that of lithium ions, the radius of sodium ions (1.02 Å) is larger than that of lithium ions (0.76 Å), which poses a challenge for the rapid insertion and extraction of Na$^+$ in the host material, which leads to a lower reversible capacity and inferior cycling performance [4–6].

Graphite, commonly used as the anode in commercial lithium-ion batteries with a layer spacing of 0.3354 nm, is not suited for the injection of sodium ions, leading to a poor cyclic stability and limited reversible capacity [7–9]. In addition to carbon-based materials, many materials for anodes, including oxides, sulfides, and selenides, have been studied for SIBs. Specifically, transition-metal phosphides (TMPs) (M=Co, Fe, Ni, Cu) [10–13], known for their high reversible capacity, have arisen as a promising choice for use in anodes for sodium-ion storage [4,14,15]. However, metallic phosphides possess a low electrical conductivity and a large volume expansion during repeated sodiation/desodiation cycles, which pose challenges to the kinetic performance. To overcome these limitations, the incorporation of nanostructured metallic phosphides into a carbon framework has proven to be a promising strategy [16–19].

In recent years, the field of battery research has shown great interest in heterostructures composed of bimetallic compounds with varying band gaps. For instance, Yue and colleagues prepared $Sb_2S_3$@SnS@C nanocomposites with hollow-tube heterostructures and achieved a capacity of 448 mA h g$^{-1}$ in SIBs at 5.0 A g$^{-1}$ by utilizing the internal electric field resulting from the heterojunction of $Sb_2S_3$ and SnS [20]. Mai et al. introduced ZIF-8 into $WS_2$ nanorods, creating abundant sulfur vacancies and $WS_2$/ZnS heterojunctions; after 5000 cycles at a high rate of 5 A g$^{-1}$, their material showed a reversible capacity of 170.8 mAh g$^{-1}$ [21]. This heterogeneous structure consists of two nanocrystalline pairs with different band gaps, which can stimulate built-in electric field effects that promote rapid charge transport and favorable reaction kinetics. Ma et al. reported the development of three-dimensional (3D) FeP/CoP heterostructures in nitrogen-atom-doped carbon aerogels (FeP/CoP-NA). The FeP/CoP-NA structure demonstrated a high-rate capability (342 mA h g$^{-1}$ at 5 A g$^{-1}$), highlighting the excellent synergistic effect between FeP and CoP through density functional theory calculations (DFT) and electro-chemical analyses [22].

Inspired by these remarkable works, we prepared bimetallic FeP/C@CoP composites using a two-step solvothermal method in this study. Firstly, cobalt-based MOFs were coated onto MIL-88 through an in situ growth method. Then, the structure of the FeP/C coating with CoP was obtained as a composite via the calcination and phosphating method. This well-crafted structure promotes rapid Na$^+$ diffusion and accelerates charge transfer at the heterogeneous interfaces. Furthermore, the robust synergistic coupling between the internal FeP and external CoP results in a stable nanostructure, greatly expediting electron/ion transport and effectively mitigating volume expansion during extended cyclic testing. Consequently, the FeP/C@CoP composite material, employed as an anode in sodium-ion batteries, displayed a capacity of 321.9 mA h g$^{-1}$ after 800 cycles at 1 A g$^{-1}$. This designed anode material shows excellent reversible capacity and stability under long cycling and could be an option for sodium-ion batteries.

## 2. Experimental Section

### 2.1. Materials Preparation

#### 2.1.1. Synthesis of Fe-MOF (MIL-88)

The synthesis of the Fe-MOF (MIL-88) followed a standard solvothermal process [23]. Firstly, $Fe(NO_3)_3 \cdot H_2O$ (1.28 g) and fumaric acid (0.336 g) were dissolved in 64 mL of N,N-dimethylformamide (DMF) to form a clear solution. After stirring for 20 min, the solution was transferred into an 80 mL stainless-steel Teflon-lined autoclave and held at 110 °C for 1 h. The resulting precipitate was washed using DMF and ethanol and collected through centrifugation. This was repeated three times, and the product was then dried at a temperature of 80 °C for 12 h in a vacuum oven.

#### 2.1.2. Synthesis of $Co_3Fe_7$/C

In a typical synthesis, 50 mg of the Fe-MOFs was dispersed in a solvent mixture consisting of 2 mL of methanol ($CH_3OH$) and 14 mL of N,N-dimethylformamide (DMF), followed by one hour of sonication to ensure even dispersion. Subsequently, 58 mg of $Co(NO_3)_2 \cdot 6H_2O$ (equivalent to 0.2 mmol) and 16.4 mg of 2-methylimidazole (equivalent to 0.2 mmol) were added to the solution. After vigorous stirring for 30 min, the solution was transferred into a 50 mL stainless-steel Teflon-lined autoclave and placed in a drying oven at 85 °C for 72 h. The resulting precipitates were collected via centrifugation, washed with deionized water and ethanol, and then dried at 80 °C for 10 h under vacuum to obtain the Fe-MOF@Co-based nanoarray composites [24]. Finally, the samples were calcined at 500 °C for 3 h under a flowing argon atmosphere.

#### 2.1.3. Synthesis of FeP/C@CoP

The FeP/C@CoP composites were synthesized through the phosphidation of $Co_3Fe_7$/C. In a typical procedure, $Co_3Fe_7$/C and $NaH_2PO_2$ were placed in separate compartments within a sealed porcelain crucible, with the $NaH_2PO_2$ located upstream. The weight ratio

of $Co_3Fe_7/C$ to $NaH_2PO_2$ was 1:17. Afterwards, the samples were calcinated at 350 °C for 2 h at a heating rate of 5 °C per minute in a static argon atmosphere. After cooling to a suitable temperature under a stream of argon, the black FeP/C@CoP material was obtained [25]. The FeP/C composites were obtained in a similar manner, with the exception of the Co-based nanoarray growth step.

### 2.1.4. Synthesis of CoP/C

To synthesize the CoP/C, 498 mg of $Co(NO_3)_2 \cdot 6H_2O$ and 656 mg of 2-methylimidazole were dissolved in 100 mL of methanol to create a homogeneous solution. This solution was left to stand for 24 h at room temperature. It was then centrifuged and washed three times with methanol to obtain ZIF-67. The resulting ZIF-67 samples were subsequently carbonized and phosphated under the same conditions as the FeP/C@CoP.

### 2.2. Materials Characterization

The structure of samples was analyzed using an X-ray diffractometer (XRD, Bruker, Ettlingen, Germany, Cu K$\alpha$ radiation). Scanning electron microscopy (SEM, Hitachi SU-70, Tokyo, Japan) and transmission electron microscopy (TEM, TECNAI-F30, Philips-FEI, Eindhoven, The Netherlands) were used to study the morphology and structure. X-ray photoelectron spectroscopy (XPS) data were collected using EscaLab250Xi (Thermo Scientific, Waltham, MA, USA). The graphitization of the material was examined by Raman spectroscopy (Jobin-Yvon, Paris, France). Nitrogen adsorption/desorption isotherms were generated using a TriStar 3020 system. Thermogravimetric analysis (TGA) was conducted using an SDT-Q600 thermal analyzer (TA Instruments, New Castle, DE, USA) in an air atmosphere.

### 2.3. Electrochemical Measurements

By mixing the FeP/C@CoP composite powder, Super P, and sodium carboxymethyl cellulose (CMC) in a mass ratio of 7:2:1 and adding water as solvent, a slurry was obtained. The slurry was then coated onto a copper foil and then dried overnight. The mass of the sample loaded on a single electrode approached 1.0 mg cm$^{-2}$. The cell assembly was operated in a glove box. Sodium tablets were employed as the counter-electrode, and a glass-fiber diaphragm (Whatman, Maidstone, UK, GF/D) was used as a separator. The electrolyte used was $NaClO_4$ (1 M) in a mixture of EC/DMC (1/1, by volume) with 5% FEC, creating 2032 button batteries.

Electrochemical performance tests of these cells were carried out in the voltage range from 0.01 to 2.5 V using a battery test system (LAND CT2001A, Wuhan, China). Cyclic voltammetry (CV) and electrochemical impedance spectroscopy (EIS) measurements were recorded using an electrochemical workstation (CHI660E, CH Instruments, Shanghai, China). A full cell was assembled with commercial $Na_3V_2(PO_4)_3$ as the cathode and FeP/C@CoP as the anode in 1 M $NaClO_4$ in EC:DEC (volume ratio 1:1) with 5% FEC as the electrolyte. The cathode-to-anode mass ratio was approximately 3.2:1.

## 3. Results and Discussion

Bimetallic phosphated FeP/C@CoP composites were synthesized using a two-step solvothermal method. As illustrated in Figure 1, the process commenced with the preparation of the Fe MOF (MIL-88B) precursor via a solvothermal approach. Subsequently, the growth of the cobalt-based nanoarray on the Fe MOF also took place through a solvothermal method. In the high-temperature solvent, a portion of the $Co^{2+}$ infiltrated the interior through the porous structure of the Fe-MOF. Simultaneously, another segment of the $Co^{2+}$ utilized the abundant functional groups on the Fe-MOF's surface as nucleation sites, binding with 2-methylimidazole to create the Co-based nanoarray. During the high-temperature carbonization stage, the $Fe^{2+}$ and $Co^{2+}$ were embedded within the carbon matrix uniformly, forming $Co_3Fe_7/C$ composites. The phosphating process led to the conversion of $Co_3Fe_7/C$ into FeP/C@CoP composites, whereby, with rising temperature, the $NaH_2PO_2 \cdot H_2O$ decomposed into $PH_3$, which reacted with $Co_3Fe$ to form FeP/CoP heterostructures [22].

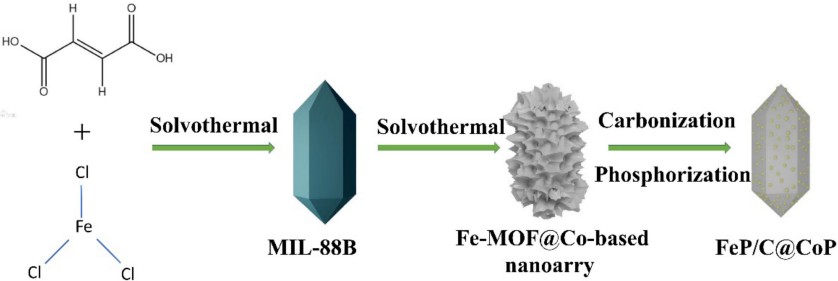

**Figure 1.** Schematic process of the synthesis process of the FeP/C@CoP composites.

The internal structural characteristics and typical morphologies of the synthesized samples were revealed using transmission electron microscopy (TEM) and field emission scanning electron microscopy (FESEM). As depicted in Figure 2a, the prepared Fe-MOF exhibited a spindle shape with good uniformity, measuring approximately 500 nm in width and 1 μm in length. The Co-based nanoarray was uniformly grown on the surface of the Fe-MOF (Figure 2b) through a solution heat method. After phosphating, the surface of the samples became rough, and some nanoparticles appeared (Figure 2c) due to the aggregation of transition metal phosphide, which was aimed at reducing the surface energy during the phosphating process [26].

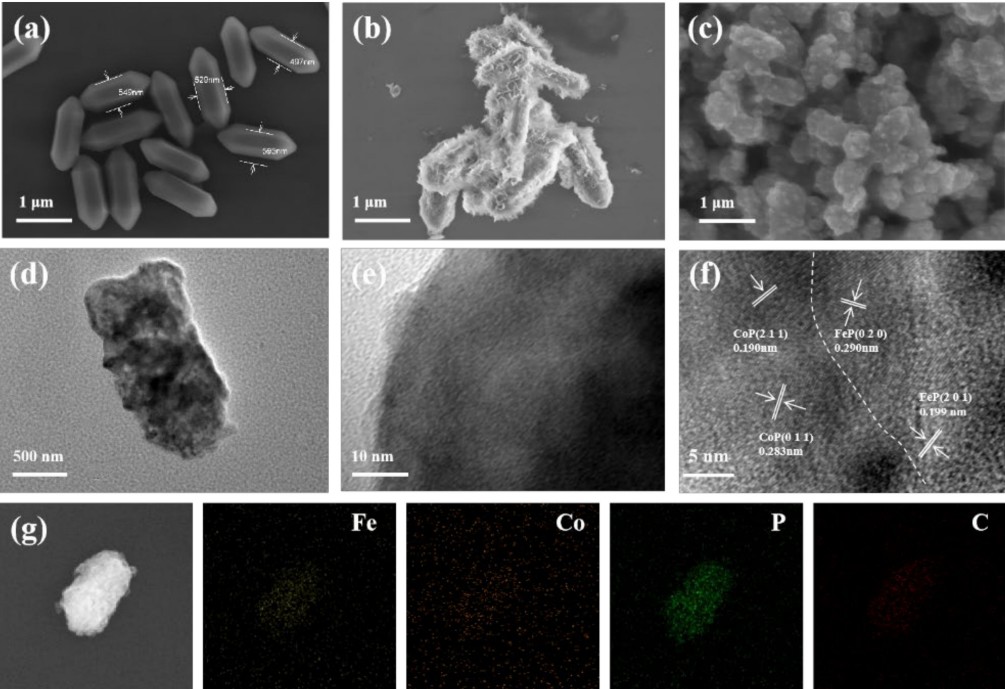

**Figure 2.** SEM images of (**a**) Fe-MOF (MIL-88), (**b**) Fe-MOF@Co-based nanoarray, (**c**) FeP/C@CoP composites; (**d**) TEM, (**e**,**f**) HRTEM, and (**g**) elemental mapping images of FeP/C@CoP.

In the TEM images of the FeP/C (Figure S2b,c) and FeP/C@CoP (Figure 2d,e) composites, it is evident that the transition metal phosphide particles were evenly embedded in the carbon matrix. This integration served to alleviate the volume expansion of the active substance during the charge and discharge cycles while enhancing the conductivity. The high-resolution TEM images (Figure 2f) revealed different lattice spacings, confirming the presence of distinct species. The lattice fringes with d = 0.190 nm and d = 0.283 nm corresponded to the (2 1 1) and (0 1 1) crystal planes of CoP, respectively, while the lattice fringes with d = 0.290 nm and d = 0.199 nm corresponded to the (0 2 0) and (2 0 1) crystal planes of FeP. Notably, the lattice angle mismatch and clearly visible lattice boundaries

between the FeP and CoP strongly support the phase–interface relationship. This atomic interface was expected to enhance the capacity for Na$^+$ storage. Figure 2g displays the EDS mapping image of the FeP/C@CoP composites, indicating the homogeneous distribution of the C, P, Fe, and Co elements. Furthermore, EDS mapping was employed to calculate the stoichiometric ratio of the elements, as presented in Table S1 of Supplementary Materials, demonstrating a Co-to-Fe ratio of 3:7.

XRD patterns were utilized to elucidate the phase components of the obtained samples. In Figure S1a, it can be observed that, following the carbonization of the Fe-MOF@Co-based nanoarray, the resulting product closely matched the crystal structure of $Co_3Fe_7$ (PDF No. 48-1817). This correspondence strongly suggested that the Fe and Co elements had undergone reduction, forming a $Co_3Fe_7$ alloy during the carbonization process. This further substantiated the formation of a heterostructure within the composite. Turning to the XRD pattern of the FeP/C@CoP composites (Figure 3a), we discerned peaks at 31.8°, 46.0°, and 48.4° corresponding to the (0 1 1), (2 0 2), and (1 0 3) planes of FeP (JCPDS No: 39-0809) [4], while the other diffraction peaks at 34.9°, 36.5°, and 57.0° represented the (200), (102), and (212) planes of CoP (JCPDS No: 29-0497) [16]. These XRD results provide compelling evidence of the successful formation of the FeP/C@CoP nanorod composites through the phosphorization treatment applied to the Fe-MOF@Co-based nanoarrays [27].

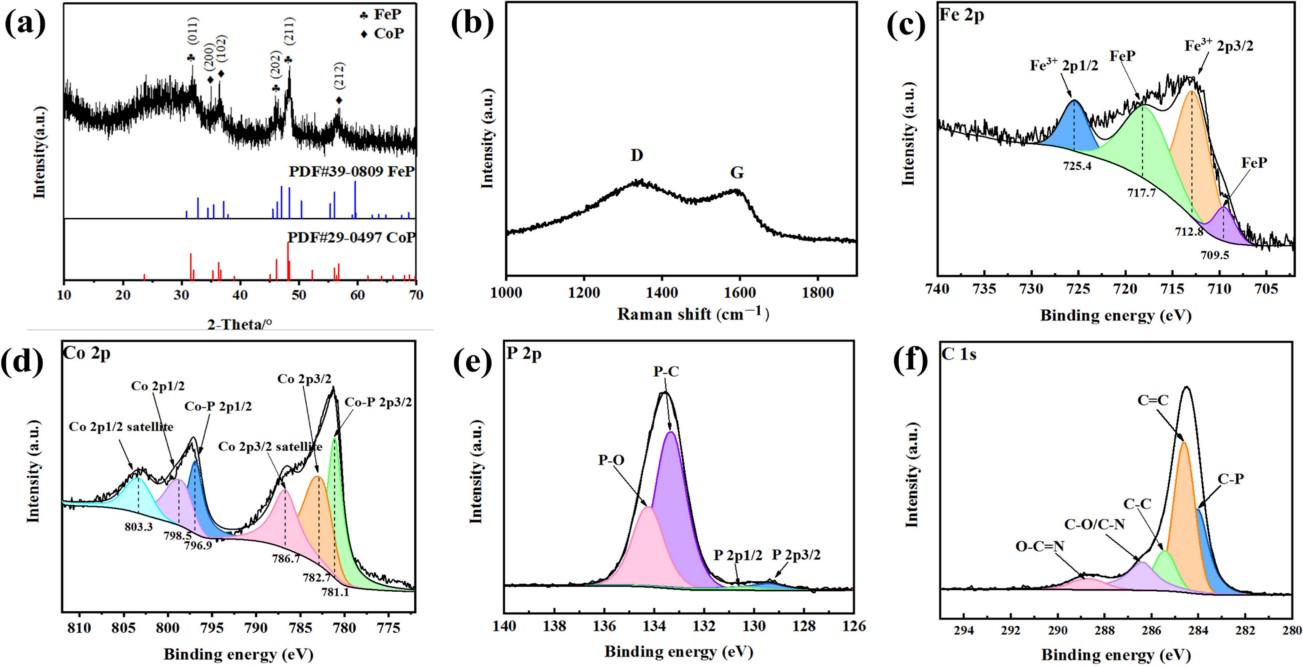

**Figure 3.** (**a**) X-ray diffraction pattern, (**b**) Raman spectrum, and (**c**–**f**) high-resolution XPS spectra of FeP/C@CoP.

Moving to the Raman spectra depicted in Figure 3b, two distinctive peaks, D-band and G-band, were evident at 1332 and 1592 cm$^{-1}$, respectively [28]. The relative strength of $I_D/I_G$ indicates the ratio of disordered carbon to graphitic carbon. In this case, the Raman tests indicated an $I_D/I_G$ value of 1.12, signifying that the carbon exhibited amorphous properties and possessed more defects. This characteristic is significant, as it offers a higher density of active reaction sites and enhanced conductivity.

For a deeper understanding of the near-surface chemical states of the FeP/C@CoP nanorods, XPS analysis was applied. Figure S3 illustrates the survey spectra of the three samples, revealing the homogeneous distribution of C, P, Fe, and Co elements in the FeP/C@CoP composites, according to the results of the elemental mapping. The presence of oxygen can be attributed to surface oxidation or the oxidation process.

The high-resolution C1s spectrum exhibited five discernible peaks located at binding energies of 284.08, 284.62, 285.44, 286.40, and 288.73 eV (Figure 3f). These peaks corresponded to different chemical bonds, i.e., C-C, C=C, C-C, C-O/C-N, and O-C=O bonds, as elucidated in the literature [29–32]. The Fe 2p and Co 2p spectra of FeP/C@CoP are displayed in Figures 3c and 3d, respectively. In the Co 2p spectrum, two principal peaks can be observed at 781.1 and 796.9 eV, belonging to Co $2p_{3/2}$ and Co $2p_{1/2}$. Additionally, other peaks centered at 782.68, 798.47, 786.70, and 803.32 eV were attributed to Co $2p_{3/4}$, Co $2P_{1/2}$ Co $2p_{3/2}$ satellite, and Co $2P_{1/2}$ Satellite, respectively (Figure 3d), as previously reported [33–35]. In the Fe 2p spectrum, the peaks located at 712.8 and 725.4 eV corresponded to Fe $2p_{3/2}$ and Fe $2p_{1/2}$ orbitals. Notably, the peaks observed at 709.5 and 717.7 eV were attributed to Fe-P in the FeP/C@CoP composition, in line with earlier studies [36]. The high-resolution P 2p results are presented in Figure 3e, wherein the peaks at 129.4, 130.4, and 133.3 eV belonged to P $2p_{3/2}$, P $2p_{1/2}$, and P-C. An additional peak at 134.2 eV was ascribed to P-O, resulting from the oxidation of phosphorus under air conditions [16,37].

The electrochemical performance of the FeP/C@CoP composites was systematically examined in coin cells (CR2032), employing sodium as the counter electrode. Cyclic voltammogram (CV) curves of the composites were utilized to probe the detailed electrochemical behavior during the $Na^+$ insertion/extraction processes and were conducted at a scan rate of 0.1 mV $s^{-1}$ within the range of voltage from 3.0 to 0.01 V vs. $Na^+$/Na. As illustrated in Figure 4a, during the first cathodic scan, a broad peak was observed at 0.5 V, corresponding to the sodium-ion insertion into the CoP and FeP [38], along with the irreversible reaction associated with SEI formation [27]. In subsequent sweeps, a noticeable decrease in the peak intensity and integral area indicated an irreversible capacity, attributed to SEI formation. The primary cathodic peak shifted to 1.0 V, a common phenomenon in LIBs and SIBs resulting from substantial structural or textural modifications during the process of initial discharge. During the anodic scan, another broad peak at 1.5–2.2 V corresponded to the sodium-ion extraction from the matrix. The CV profiles demonstrated the good electrochemical reversibility of the electrode materials during the discharge–charge cycles, particularly at the second and thi3rd cycles. According to the CV results and the analysis of the related literature [39], the charge and discharge process of FeP/C@CoP can be depicted as follows:

$$\text{FeP(CoP)} + 3Na^+ + 3e^- \rightarrow \text{Fe(Co)} + Na_3P \text{ (initial discharge)} \tag{1}$$

$$Na_3P \leftrightarrow 3Na^+ + 3e^- + P \tag{2}$$

The first three volume–voltage curves of the FeP/C@CoP are displayed in Figure 4b at a current density of 0.1 A $g^{-1}$ within the voltage range of 0–3.0 V. In the first cycle, the discharge and charge capacities of the FeP/C@CoP anode amounted to 723.7 and 400.9 mA h $g^{-1}$, respectively, with an initial coulombic efficiency of 55.3%. Notably, the voltage plateau in the volume–voltage profile aligned with the CV curve. The subsequent two cycles largely overlapped, signifying the material's strong cyclic stability. In contrast, the initial discharge/charge specific capacities of the FeP NPs electrodes were 662.4 and 319.1 mA h $g^{-1}$, respectively, resulting in an initial Coulomb efficiency (ICE) of only 48.2% (Figure S4). Figure 4c shows the comparison of the cycling performance of the FeP/C@CoP and FeP/C electrodes at a current density of 0.1 A $g^{-1}$. The FeP/C@CoP electrode maintained a high specific capacity of 416 mA h $g^{-1}$ over 100 cycles without significant degradation. For the FeP/C electrode, on the other hand, the specific capacity was only 233.5 mA h $g^{-1}$ after 100 cycles, which confirms the enhancement in the heterostructure for the electrochemical performance of the FeP/C@CoP composites. Moreover, as shown in Figure 4d, the FeP/C@CoP electrode provided specific capacities of 410.3, 382.6, 349.1, 335.8, 314.2, and 275.7 mA h $g^{-1}$ at 0.1, 0.2, 0.5, 1, 2, and 5 A $g^{-1}$ current densities, respectively, while the FeP/C electrode showed much lower specific capacities of 261.2, 220.7, 190.0, 167.6, 150.1, and 112.3 mA h $g^{-1}$ at the same current densities. The capacity retention ratio of the FeP/C@CoP and FeP/C electrodes at 5 A $g^{-1}$ compared to

$0.1 \text{ A g}^{-1}$ was 67.2% and 43.0%, respectively, emphasizing the superior rate performance of the FeP/C@CoP. The substantial increase in the rate capacity of the FeP/C@CoP was due to the formation of heterostructures. Notably, Figure 4e highlights the remarkable cyclic stability of the prepared FeP/C@CoP electrode with minimal capacity fluctuation. After 800 cycles at a high current of $1 \text{ A g}^{-1}$, the capacity remained at $321.9 \text{ mA h g}^{-1}$, and the Coulombic efficiency was approximately 100%. In contrast, the capacity of the FeP/C was only $134 \text{ mA h g}^{-1}$. To further demonstrate the performance enhancement of the heterostructure in the FeP/C@CoP, the FeP/C and CoP/C samples were mechanically mixed in the atomic ratio (FeP/C:CoP/C = 7:3), and relevant electrochemical tests were performed for comparison. Clearly, the CV curves (Figure S5a,b) under the same conditions as the FeP/C@CoP exhibited similarities, indicating a similar charge–discharge mechanism of FeP and CoP. As shown in Figure S5c–e, the capacity, rate performance, and cycling stability of the FeP/C@CoP electrodes outperformed those of the CoP/C and FeP/C. These results underscore that the performance improvement in the FeP/C@CoP material, as compared to FeP/C, was not simply due to the addition of CoP but was a result of the superior structural design of the FeP/C@CoP material and the formation of a heterostructure between the FeP and CoP.

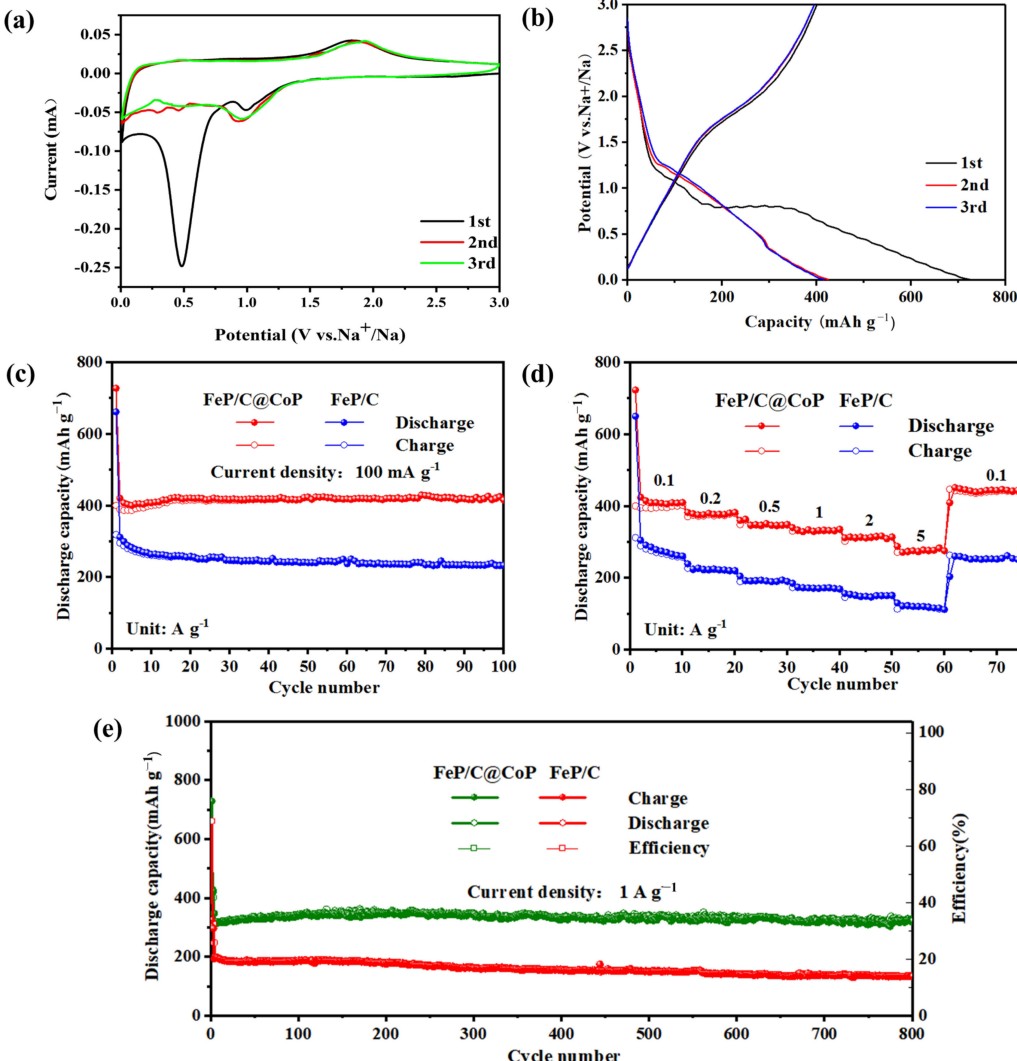

**Figure 4.** Storage performance tests of FeP/C@CoP composites: (**a**) the first three CV curves at $0.1 \text{ mv s}^{-1}$ at 0.01–3.0 V; (**b**) the first three capacity–voltage curves at $0.1 \text{ A g}^{-1}$; (**c**) cycling performance of FeP/C@CoP, FeP/C at current of $0.1 \text{ A g}^{-1}$; (**d**) the rate performance from $0.1 \text{ A g}^{-1}$ to $5 \text{ A g}^{-1}$; (**e**) long-cycle performance at $1 \text{ A g}^{-1}$.

Electrochemical impedance spectroscopy (EIS) measurements were performed in order to clarify the reason for the excellent sodium storage performance of the FeP/C@CoP electrodes [40,41]. The Nyquist diagram of the FeP/C@CoP and FeP/C electrodes prior to cycling is illustrated in Figure 5. Notably, the charge transfer resistance of the FeP/C@CoP electrode ($R_{ct}$ = 638 ohms) was lower than that of the FeP/C electrode ($R_{ct}$ = 834 ohms). To delve deeper into the electrochemical reaction kinetics of the FeP/C@CoP, CV tests were performed on the FeP/C@CoP (Figure 6a) and FeP/C (Figure S6) electrodes at scan rates ranging from 0.2 to 1.0 mV$^{-1}$. This allowed for a more comprehensive investigation of the reasons behind the superior sodium-ion storage capacity of the FeP/C@CoP anode. The observed response kinetics encompassed both diffusion and pseudocapacitive behavior during the battery charging and discharging processes. As per the derivation of the power law [42–44]:

$$i = a \times V^b \tag{3}$$

where both *a* and *b* are constants, with the b value determined by the slope of log (*i*) versus log (*v*). The calculated *b* value was 0.5, indicative of the diffusion-rate-determining process. In contrast, *a b* value of 1 signifies the domination of capacitive behavior [45–47]. In Figure 6c, the *b* values for peak 1 and peak 2 were fitted to 0.841 and 0.818, respectively, suggesting that the kinetics of the FeP/C@CoP are primarily governed by surface capacitance.

$$i(V) = k_1 v + k_2 v^{1/2} \tag{4}$$

$k_1$ and $k_2$ indicate the corresponding contribution ratio of capacitance and diffusion control behavior in the above equation [48]. At a scan rate of 1 mV$^{-1}$, the pseudocapacitive behavior contribution of the FeP/C@CoP electrode was 87.4%, higher than that of the FeP/C electrode, which exhibited an 82.9% pseudocapacitive behavior. Additionally, the pseudocapacitive behavior contribution of the FeP/C@CoP electrode exceeded that of the FeP/C electrode at different scan rates, reaffirming the positive impact of the heterostructure formation in the FeP/C@CoP on the capacitance contribution.

Considering the remarkable performance of the FeP/C@CoP in half batteries, we proceeded to assemble a full SIB cell with an FeP/C@CoP anode and a $Na_3V_2(PO_4)_3$ (NVP, theoretical capacity 118 mA h g$^{-1}$) cathode. In Figure 7a, the galvanostatic discharge/charge profiles of the NVP//FeP/C@CoP full cell are presented. The full cell underwent cycling within the voltage range of 1.0–3.7 V, demonstrating an average performance. Notably, the FeP/C@CoP electrode achieved a remarkable charge/discharge capacity of 528.6/154.7 mA h g$^{-1}$. Even more impressive is the sustained capacity of 416 mA h g$^{-1}$ after 100 cycles at a current density of 100 mA g$^{-1}$, underscoring its exceptional cycle stability. This outstanding electrochemical performance in the full cell suggests that FeP/C@CoP composites hold immense potential for high-performance SIBs.

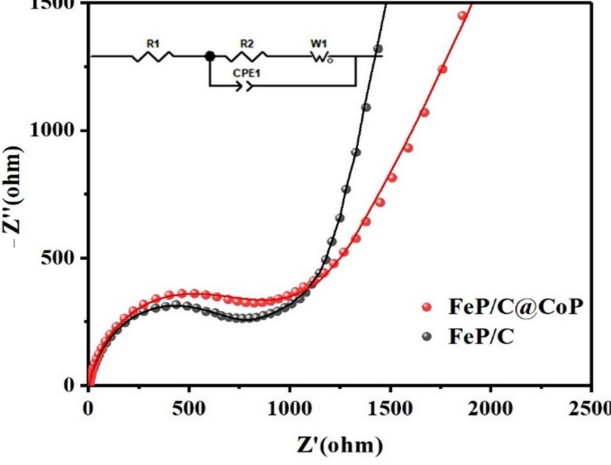

**Figure 5.** Electrochemical impedance spectroscopy of FeP/C@CoP, FeP/C electrodes.

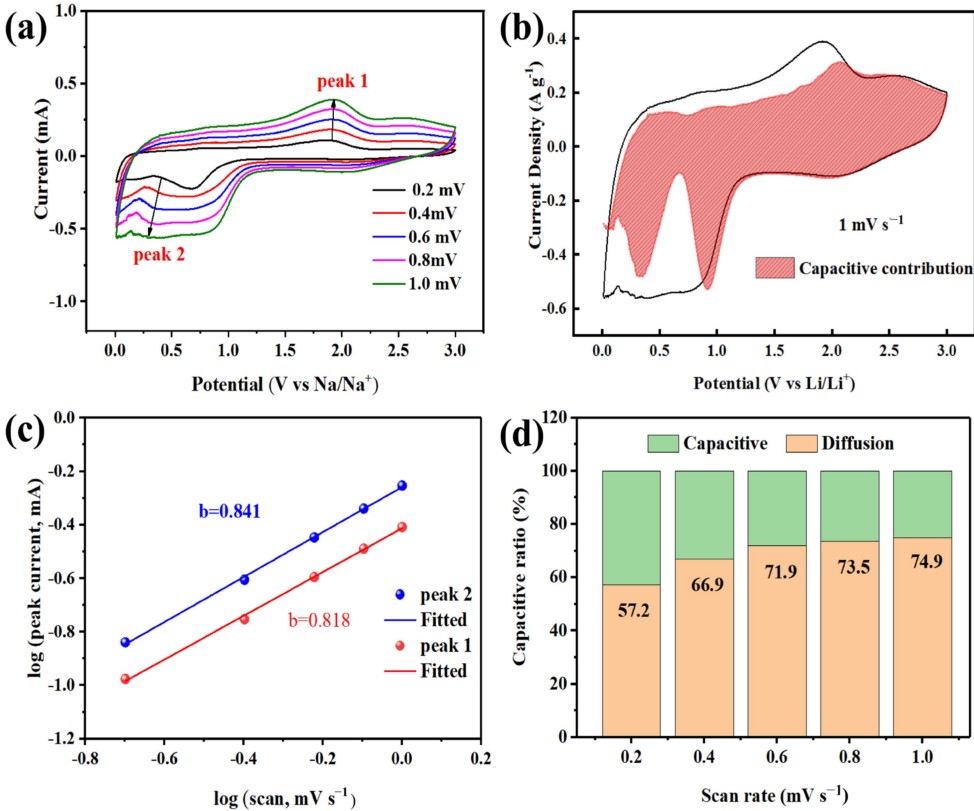

**Figure 6.** (**a**) CV curves at different scan rates, (**b**) CV curve with capacitive contribution shaded area at scan rate of 0.2 msV s$^{-1}$, (**c**) the b value graph, and (**d**) the contribution rate of capacitance.

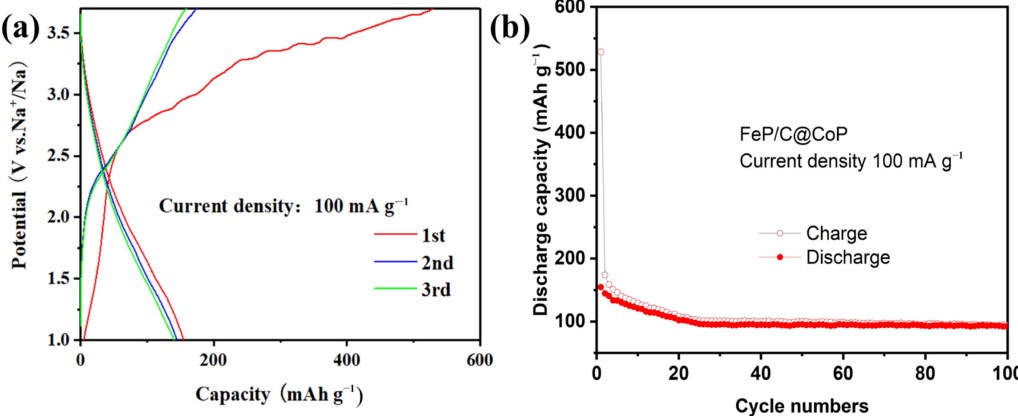

**Figure 7.** (**a**) Charge and discharge curves of the first three cycles of $Na_3V_2(PO_4)_3$//FeP/C@CoP full cell between 1.0–3.7 V. (**b**) Cycling performance of $Na_3V_2(PO_4)_3$//FeP/C@CoP full cell at a current density of 0.1 A g$^{-1}$.

## 4. Conclusions

In this study, we proposed a method for the facile synthesis of MOF coated on the outer layer of MOF. The heterostructures obtained improved the electrochemical kinetics and exhibited excellent revertible capacity when used as anodes for sodium-ion batteries. The generation of FeP/C@CoP heterostructures and the resulting bimetallic phosphide nanoparticles embedded in the porous carbon substrates provided a lot of reaction sites and shorter Na$^+$ diffusion paths, both of which contributed to the improvement in the specific capacity and the accelerated reaction kinetics. Moreover, the establishment of a perfect atomic interface between the bimetallic phosphides also maximized the synergistic effect. As a result, the FeP/C@CoP exhibited and excellent rate capability, i.e., 416 mA h g$^{-1}$ at

0.1 A g$^{-1}$ and 321.9 mA h g$^{-1}$ at 1 A g$^{-1}$. In addition, the dynamic analysis of the cyclic voltammetry curves at different scanning rates revealed a significant pseudocapacitive behavior of the FeP/C@CoP composites during the charge/discharge process, with an apparent pseudocapacitive behavior at a scanning rate of 1 mv s$^{-1}$, where the pseudocapacitance was as high as 87.4%, which effectively improved the high current rate performance of FeP/C@CoP. This method of preparing stable and high-capacity heterostructured electrodes can be popularized in the field of sodium-ion battery anode preparation.

**Supplementary Materials:** The following supporting information can be downloaded at: https://www.mdpi.com/article/10.3390/coatings13122056/s1, Figure S1. XRD diffraction pattern of (a) Fe-MOF@Co-based nanoarry composite carbide product Co3Fe7/C; (b) Fe-MOF Phosphating product FeP/C; (c) ZIF-67 phosphating product CoP/C. Figure S2. (a) SEM image and (b,c) TEM images of FeP/C. Figure S3. XPS survey spectrum of FeP/C@CoP. Table S1. The mass fraction of an element of FeP/C@CoP. Figure S4. The first three capacity-voltage profiles at 0.1 A g$^{-1}$ of FeP/C. Figure S5. The first three CV curves at 0.1 mV s$^{-1}$ in the voltage of 0.01–3.0 V of (a) FeP/C and (b) FeP/C:CoP/C = 7:3 composites. (c) Cycling performance of FeP/C@CoP, CoP/C and FeP/C:CoP/C = 7:3 electrodes at current of 0.1 A g$^{-1}$; (d) the rate performance from 0.1 A g$^{-1}$ to 5 A g$^{-1}$; (e) the long cycling performance at 1 A g$^{-1}$. Figure S6. (a) The CV curve of the FeP/C electrode at different scan rates from 0.1 to 1 mV s$^{-1}$. (b) the pseudo-capacitance contribution at 1 mV s$^{-1}$. (c) the b-value graph of the two curves. (d) the contribution rate of capacitance at different scan rates.

**Author Contributions:** Conceptualization, Z.Z.; Validation, Z.Z.; Formal analysis, H.D.; Investigation, T.M. and Y.X.; Data curation, Z.H.; Writing—original draft, J.L.; Writing—review & editing, J.L. and G.Y.; Supervision, G.Y.; Funding acquisition, G.Y. All authors have read and agreed to the published version of the manuscript.

**Funding:** This research was funded by the National Natural Science Foundation of China (grant No. 51971184) and the Natural Science Foundation of Fujian Province of China (No. 2021J01043 and No. 2023J01033).

**Institutional Review Board Statement:** Not applicable.

**Informed Consent Statement:** Not applicable.

**Data Availability Statement:** All relevant data are within the paper.

**Conflicts of Interest:** The authors declare no conflict of interest.

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
