# Peer review of "Facilitating Synthesis of FeP/C@CoP Composites as High-Performance Anode Materials for Sodium-Ion Batteries"

_coatings, doi:10.3390/coatings13122056_

Round 1

Reviewer 1 Report

Comments and Suggestions for Authors

Author Response

Responds to the reviewers’ comments:

Reviewer:  

The proposed paper is under consideration for publication in this journal after minor revisions. Authors are kindly requested to consider the following comments:

Q1/ Underline the novelty of this work

Response: Thanks for your kind suggestions. We mark the novelty in the manuscript.

Q2 / Give the name of each composite in the abstract (e.g. FeP/C@CoP, Co-ZIFs.......)

The abstract must be attractive to show the interest of this study.

Response: Thanks for your kind suggestions. Due to the complex composition of the synthesized compounds and the lack of suitable names, we refer to the composites by abbreviated names for ease of understanding.

Q3/ If possible, give the reference of all the synthesis procedures.

Response: Thanks for your kind suggestions. We added the references where need.

Q4/ For greater clarity, you should add a scheme showing the electrode manufacturing procedure (graphical abstract).

Response: Thanks for your kind suggestions. We make the electrode through a simple and general procedure. Firstly, we prepared a slurry with the synthesized composites, then hand-coated onto a copper foil and dry overnight.

Q5/ The mechanism proposed on pages 3 and 4 is not referenced.

Response: Thanks for your kind suggestions. We added the references.

Q6/ The Miller indices and the corresponding crystal planes are not referenced. The same applies to XRD and Raman (lines 179 and 180).

Response: Thanks for your kind suggestions. We added the references.

Q7/ Table 1 is missing.

Response: Thanks for your kind suggestions. Table 1 is in the supporting information, we fixed it for clear.

Q8/ Improve the visibility of figures 3 and 4b.

Response: Thanks for your kind suggestions. We changed the figures 3 and 4b.

Q9) line 191: indicate the corresponding figure (fig.3 f). Ditto for line 194.....

Response: Thanks for your kind reminding. We revised it.

Q10/ Equations 1 and 2 are not balanced, please correct them and suggest an appropriate reference and indicate the oxidation and reduction potential for each reaction. In addition, the CV does not indicate any reversible reactions, so think about equation 2 in which you mention a reversible system.

Furthermore, reversibility can be confirmed by the positive shift of the Ep with increasing scan speed:

+ cf. Microchimica Acta (2021) 188 : 184

Response: Thanks for your kind suggestions. We carefully checked and revised it and added the reference.

Q11/ Diffusion/adorption can be confirmed by plotting log Ip versus logv (Fig.6c). The following reference is an interesting way of illustrating this

+ Chemical Data Collect. 31(2021), 100595.https://doi.org/10.1016/j.cdc.2020.100595

Response: Thanks for your kind suggestions. We added this reference.

Q12/ How do you calculate the contribution of the pseudocapacitive behaviour of the FeP/C@CoP electrode (87.4%)?

Response: Thanks for your kind asking. The pseudocapacitance contribution value obtained by dividing the area of the fitted curve by the area of the CV curve at a scan rate of 1 mV-1. (Figure 6b)

Q13/ Think about reducing the percentage of plagiarism (44%).

Response: Thanks for your kind suggestions. We have scrutinized and revised the entire manuscript.

Q14/ Can you give the cost of the synthesised electrode and its lifetime?

Response: Thanks for your kind suggestions. The total cost of the synthesized electrode was about 0.1$ based on the reagents we used. Its lifetime was estimated to be 2 years based on cyclic stability experiments over 800 cycles.

Reviewer 2 Report

Comments and Suggestions for Authors

Dear Editor,

The authors have proposed an anode material for high-capacity sodium-ion batteries. The topic of their research is intriguing and has practical applications. I found the manuscript to be well-organized and well-written. Considering these factors, I believe the manuscript can be accepted with minor revisions, which are as follows:

1. Please consider adding "e FeP/C@CoP Composite" to the list of keywords.

2. Could you provide information about the temperature at which the experiments were conducted? Additionally, has the effect of temperature on battery performance been considered?

3. please elaborate on the contact resistance during discharge and charge regimes.

4. Has the possible corrosion of the anode been considered over time?

 Kind regards,

Author Response

Responds to the reviewers’ comments:

Reviewer:  

  1. Please consider adding "FeP/C@CoP Composite" to the list of keywords.

Response: Thanks for your kind suggestions. This is a very good suggestion. We added it to the keywords list.

  1. Could you provide information about the temperature at which the experiments were conducted? Additionally, has the effect of temperature on battery performance been considered?

Response: Thanks for your asking. All synthesis temperatures are indicated in the experimental section. The electrochemical performance as well as cell performance test experiments are conducted at room temperature. We haven’t thought about the effect of temperature on battery performance since batteries are generally used at room temperature. We may consider it in the future.

  1. please elaborate on the contact resistance during discharge and charge regimes.

Response: Thanks for your kind suggestions. We use electrochemical impedance spectroscopy (EIS) to estimate the resistance during charging and discharging. We added the discussion in the text.

  1. Has the possible corrosion of the anode been considered over time?

Response: Thanks for your asking. The anode barely corrodes over time. We did the long cycling tests and the capacity was still 326 mA h g−1 after 800 cycles at a current density of 1 A g-1.

Reviewer 3 Report

Comments and Suggestions for Authors

In this paper, Mao et al., prepared a bimetallic composite (c FeP/C@CoP) by a two-step solvothermal method. They used this as an anode material in sodium-ion batteries. The idea is good and work conducted well. Therefore, I am supportive of its publication but after some major revisions.

-          The title of the paper is not appealing, it should be revised.

-          Please revise the abstract and state clearly how this work is important for society and what’s the novelty of this work, key techniques/methods/results, etc.

-          Please focus on the key things and add additional recent relevant details in the introduction part.

-          Most of the statements in the experimental part are without references, please add references for each used statement in the experimental section if its not new. It’s necessary for reproducibility.

-          Figure 4-part c, d and e is not explained well. Please add more comprehensive discussion relevant to these sections.

-          Most of the equations are without references, and not explained well. Please fix this issue in your whole manuscript.

-          Extensive English corrections are needed.

-          The conclusion is too generic, please add key findings there. 

Comments on the Quality of English Language

Extensive editing of English language required

Author Response

Responds to the reviewers’ comments:

Reviewer:  

  1. The title of the paper is not appealing, it should be revised.

Response: Thanks for your kind suggestions. We revised the title to “Facilitating Synthesis of FeP/C@CoP Composites as High-Performance Anode Materials for Sodium-Ion Batteries.”

  1. Please revise the abstract and state clearly how this work is important for society and what’s the novelty of this work, key techniques/methods/results, etc.

Response: Thanks for your kind suggestions. We revised the abstract and stated the importance and novelty of this work.

  1. Please focus on the key things and add additional recent relevant details in the introduction part.

Response: Thanks for your kind suggestions. We revised the introduction. Added recent research and progress of sodium-ion battery.

  1. Most of the statements in the experimental part are without references, please add references for each used statement in the experimental section if its not new. It’s necessary for reproducibility.

Response: Thanks for your kind suggestions. We added references where needed.

  1. Figure 4-part c, d and e is not explained well. Please add more comprehensive discussion relevant to these sections.

Response: Thanks for your kind suggestions. We elaborated on this part of the discussion, please see lines 244 to 260.

  1. Most of the equations are without references, and not explained well. Please fix this issue in your whole manuscript.

Response: Thanks for your kind suggestions. We fixed this and added references where needed.

  1. Extensive English corrections are needed.

Response: Thanks for your kind suggestions. We carefully checked the manuscript and made thorough corrections.

  1. The conclusion is too generic, please add key findings there.

Response: Thanks for your kind suggestions. We rewrote the conclusion.

Round 2

Reviewer 3 Report

Comments and Suggestions for Authors

Well revised-Accept

Comments on the Quality of English Language

Extensive editing of English language required